# Students' Workplace Readiness: Assessment and Skill-Building for Graduate Employability

Sara Siddique [1] , Ali Ahsan [2] , Neda Azizi [3],* and Omid Haass [4]

1 Faculty of Management Sciences, Foundation University Islamabad, Islamabad 44000, Pakistan; sara.siddique@fui.edu.pk
2 Business and Hospitality, Torrens University, Adelaide, SA 5000, Australia; al_ahsan1@yahoo.com
3 Business and Hospitality, Torrens University, Melbourne, VIC 3000, Australia
4 School of Property, Construction and Project Management, Royal Melbourne Institute of Technology University, Melbourne, VIC 3001, Australia; omid.haass@rmit.edu.au
* Correspondence: neda.azizi@torrens.edu.au

**Abstract:** This study introduces a new approach for the competence development of the socio-technical aspect. The curriculum of the Project Management (PM) course taught in degree programs concentrates largely on imparting technical knowledge. Current research seeks to direct the attention of the PM curriculum towards Personal Competencies as well. PM studies not only require a project to be carried out successfully but also expect students to demonstrate certain personal competencies, behaviors, and traits to effectively lead the project team. This research seeks to inform action and yield pertinent knowledge and instructional material around the desired personal competence. This study adopts an exploratory and (educational) action research approach with a quantitative mode of inquiry. The first of the four phases of this study comprises an action-based approach to find out which Personal Competence is essential for PM students. The second phase is of further follow-ups with the research subjects that indicate their lack of understanding of the PM personal competencies. The next phase employs a second survey showing that the Communication Skills of research subjects need work. The last phase offers a comprehensive training plan around the required competence, a training evaluation tool, a competence assessment exam, and tools for training and trainer feedback. The findings of this research bear immense implications for PM competence building and curriculum. The practical contribution of the study offers a way to prepare the graduates for ready employability.

**Keywords:** project management competency development; knowledge competence; personal competence; competency development; project management curriculum

## 1. Introduction

Technology is rapidly altering the nature of competition, and the global job market is in bigger demand for skilled and ready-for-the-job people. Getting quality education from prestigious and high-ranking colleges is normally the first step to a prosperous future. There is no denying that education and knowledge competencies play a vital role in achieving the career aspirations of the people, but does it make them industry-ready? A sizable gap exists between industry and academia. Among the many criticisms of the Higher Education Institutes (HEIs) of Pakistan, some studies establish a weak link between academia and industry [1]. Landing a job right out of college depends a great deal on the soft skills of the candidate. Personal competencies play a significant role in climbing the career ladder. Educational institutes need to groom their students, keeping in mind the industry needs before they go out into the real world. Researchers argue that the proficiency and the type of skills being developed in HEIs are not enough to match those required for employment and are mismatched to those the industry needs [2,3]. Studies with employers show that employers concur [4–6]. The rate of graduate unemployment is growing by the day, and this unemployment is leading to graduates taking on unskilled or semi-skilled roles [7,8].

One of the leading causes of graduate unemployment is graduates not meeting industry needs in terms of the skills required. Grant Thornton Consulting [9] and Uzair-ul-Hassan and Noreen [5] emphasize the employers' concerns regarding the soft skills of graduates e.g., communication skills, leadership skills, and critical thinking.

For the role of a project manager, personal and behavioral characteristics are valued as highly, if not more, as knowledge competence. An informal and unpublished study of the current Project Management (PM) curriculum being taught in tertiary education institutes at the undergraduate level shows that it is mainly being kept to develop Knowledge and Performance Competence. These competencies include Knowledge of PM, tools and techniques required to plan and manage the projects, and other applications essential to see through a project to its successful closure. Since fresh graduates, who are highly competent and industry-ready, are in demand greater than ever, this study aims to develop further the design of an effective PM curriculum and eventually good PM practices.

PM research has argued that much can still be learned from studies of PM principles and industry-wide practices, indicating a great need for Knowledge, Performance, and Personal Competencies [10]. Since Knowledge and Performance Competencies are well taken care of by academia, and Personal Competence is not being paid the attention required, action must be taken to bridge the gap between possessing the knowledge and utilizing the behaviors and core personality traits that contribute toward using that knowledge effectively. An effort is required to create knowledge around a particular personal competence for project managers for better job proficiency that shall also be relevant beyond the current research setting and on different research subjects, such as students and/or entry-level professionals, from different fields, such as electrical, civil, or mechanical engineering, biomedical or chemical engineering, etc. Competence building first calls for the identification of the lack of key personal competence. The postulate is that competence-building research is recognized as necessary to the study of a successful PM curriculum, leading to an extension of the PM literature. Students possess the knowledge but do not have or exercise the core behavioral/personality traits to effectively use that knowledge. Motivated by the gap, we address the following research question: "In which area of personal competence do undergraduate-level project management scholars lack competency, and how can the competency be built?"

## 2. A Review of Literature

### 2.1. Project Management Personal Competences

Professional competencies are vital for effective PM and for working well in the workplace setting. The main competencies required for the successful execution of a project are technical and knowledge-based. These typically relate to project process groups, such as initiation, planning, execution, monitoring and control, and closing. Competences essential for working effectively in a project team are discussed as 'soft skills,' 'people skills,' or socio-technical or managerial competencies.

Previous PM studies, unlike management literature, have largely concentrated on the success of the project and not as much on the role of project managers' managerial competencies. Effective leadership is chiefly viewed as a critical success factor of any project in general management, but the role of the project manager's competence and communication has not been discussed in the existing works on project success factors [11].

According to Boyatzis [12], competence is defined as the basic attributes of an individual that comprise subject knowledge, relevant skills, and personality traits, such as attitude, morals and values, and behavior. Competence is the condition of being amply qualified. The particular knowledge, skills, and proficiency an individual carries to their workplace define a certain competency profile. Every situation demands a unique profile. Project managers need coaching for their requisite profile to fit well into their role [13].

Professional knowledge is referred to as hard skills, whereas personal competencies and attributes are referred to as soft skills [14–16]. The Project Management Institute [17] proposes in its competence development framework that competence can fall into the three

dimensions of Knowledge, Performance, and Personal. Work readiness has always been considered an important concept [18]. Although no readily available instrument exists for the measurement of personal competence, PMI has furnished a basic framework to develop the assessment tool required to determine the current level of personal competence for project managers [17].

Munns and Bjeirmi [19] pointed out that the chief focus has always been concentrated on performance competence, with a negligible focus on behavior or emotional factors. Over two decades later, this is still true for existing literature. The International Project Management Association (IPMA) furnished a Competence Baseline v4.0 to center the attention of project managers on the requirement of their field-related knowledge and experience. The baseline addresses personality traits and behaviors and personal attitudes. However, they are not as comprehensively covered as "Practice Competence". People competence elements are defined as the personal and social competencies expected to be exhibited by the project manager and further divided into 10 elements [20]. These competencies are (1) self-reflection and self-management, (2) personal integrity and reliability, (3) personal communication, (4) relationships and engagement, (5) leadership, (6) teamwork, (7) conflict and crisis, (8) resourcefulness, (9) negotiation, and (10) result orientation.

Dulewicz and Higgs have identified 15 competencies, grouped into 3 types, Intellectual (IQ), Managerial (MQ), and Emotional (EQ), such as engaging communication, intuitiveness, interpersonal sensitivity, etc., required for effective leadership. These competencies are proposed along with three leadership styles, Involving, Engaging, and Goal-oriented, in the setting of PM. Emotional Intelligence and Communication were found to be the most considerable competencies for project managers [21].

Chen et al. [22] mentions that students grasp professional skills with relative ease compared to general competencies. Herbet et al. [23] calls for creative approaches to foster student behaviors and personality traits alongside knowledge skills. There is a lack of contemporary relevance in existing management education, and it does not necessarily speak to the employers' requirements for skills and personality traits in graduates [23]. Reedy et al. [24] concur that the incorporation of employability and personal skills with technical and knowledge skills is essential to groom industry-ready graduates. High academic performance (i.e., knowledge skills) combined with extracurricular activities (that help develop social skills and personal qualities) increase the perceived employability of business graduates in the industry [25]. Hanif et al. [26] have identified a strong need for effective training on behavioral and interpersonal skills for project directors and managers. Dubey and Tiwari [27] note a sizable gap in the soft skills possessed by the graduates and those required by the industry. A recent study confirms that employment considerations are not solely based on technical competence, but also consider soft skills [28].

In terms of competency development, PMI has had a framework in the works since 1997 to "Improve the Performance of Project Personnel." This framework first came out in 2002, and with the help of this framework, survey instruments to measure any of the competency dimensions defined therein can be constructed. The framework provides performance criteria against specific elements of competence that need to be displayed by the individuals to be considered competent [17]. As a result, it appears that the latest (third) version of the Project Management Competency Development Framework (PMCDF), released in 2017, offers a comprehensive list of elements describing what a competent person in PM should know, do, and be.

Thus, existing studies show that industry calls for graduate work-readiness and employability and puts a great deal of importance on personal competence. Now that relevance and need for personal competence in the industry have been established, the next section discusses the importance given to personal competence in academia.

### 2.2. Project Management Curricula

In HEIs, the curriculum of PM studies concentrates almost entirely on the knowledge and/or performance competence. Personal competence is often neglected and considered

not as important, resulting in students not being prepared for the job market. Personal Competencies are required for their development as well as for better job performance. Increasing demand for competent professionals exists in the field of PM, making it essential to design PM education according to industry requirements [29]. Little research has been done on graduate employability in the PM profession [30]. Ekstedt [31] also highlights the role of HEIs in preparing graduates for industry. It is incumbent on HEIs to ensure that graduates possess the necessary competencies required for graduate employability [32].

Current PM degree and training programs make evident that they are developed with a focus on standardization of the field and to prepare the project managers to deal with complexity [33]. The educational institutes mainly stress technical knowledge instead of subjective learning. However, PM in totality includes an all-inclusive understanding of personality and behavioral qualities that bear key importance for the organizations [34].

Professional PM knowledge has been considered important for decades; however, current PM education still falls short in imparting aspiring project managers with the personal and managerial competencies necessary to manage complex projects. Ample research has been conducted in PM that every take offers a distinctive outlook on this discipline. Since international recognition, PM knowledge is being formally imparted as a part of other specialized fields, as well as professional degrees. As the profession advances, PM practices have acknowledged the importance of "Human Skills". The current PM curriculum largely stresses knowledge and technical competence, i.e., hard skills, at the cost of personal competence, i.e., soft skills [29]. A recent study established that what higher education institutes offer falls short of what the industry requires from graduates [35].

Workplaces demand higher-order contemporary skills and abilities, especially those of communication, such as critical thinking, collaboration, creativity, etc. [36]. PMI [10] reports a widening gap between employers' demand for skilled PM workers and the availability of professionals for these roles. A recent study suggests that while students consider technical knowledge to be more important, employers deem soft skills, such as communication and interpersonal skills, more important [37].

Thus, in the extant literature, the focus is mainly placed on technical/knowledge competence. However, industry and PM practitioners acknowledge the importance of personal competence. A PM course demands to be designed such that together with technical knowledge, it must also introduce instructional blocks for personal competence. Alshammari et al. [38] find that the Project Manager Skills Framework should be developed to help academic and industry professionals identify the gaps in required and possessed competence. This poses the requirement for the inclusion of personal competence in the PM curriculum. This raises the question as to what personal competency aspects must be incorporated in the current PM curriculum and how.

### 2.3. Competence Building Using Bloom's Taxonomy

To design and develop the curriculum of a subject, its educational goals, and learning objectives, Bloom's Taxonomy (BT) is thought to be an all-encompassing model. BT has three domains: cognitive, affective, and psychomotor. Cognitive domain classes thinking and learning processes in a six-layer model ordered in increasing difficulty and explicit nature of learning activities. The first three layers that are considered concrete include knowledge, comprehension, application, and the latter three layers that are considered abstract include analysis, synthesis, and evaluation. To move to a higher level of the cognitive domain, the level before must be mastered. The model sees to it that activities selected or designed for a subject meet the Learning Outcomes (LOs) defined for it [39,40]. In 2001, Revised BT showed a change in its structure to a wheel-form from previously ordered layers. The nomenclature of the layers has also changed into verb form from the noun. The last two layers of "creating" and "evaluating" also switched places [41].

Studies provide ample proof that BT is a significant tool in curriculum development and bears a great impact on education. BT not being bound to a specific field offers a meticulous framework to design LOs and support instruction material and assessments [42].

Students' performance in courses planned using BT showed a higher average score compared to those planned using typical textbook-bound instruction [43]. The literature has stressed universities to ensure that graduates are prepared for project-based dynamic work environments [30] and programs are aligned with the current needs of the industry [44], and it finds that appropriate instruction material and assessment tools need to be planned and prepared to effectively develop higher-order skills. Bloom's Taxonomy of Cognitive Domain is valuable in designing instruction and preparing assessments [45]. Numerous instances exist demonstrating the usefulness of Bloom's Taxonomy for preparing tailor-made LOs and designing and selecting content for instruction and evaluation. Ching et al. [46] report improvement in average scores of students after they were exposed to competence development using Bloom's Taxonomy. Although BT is not specific to any subject, the existing literature fails to deliver any work concerning its application for developing a PM curriculum.

In sum, the review of the literature has established that the focus of industry and academia needs to be aligned. The literature also shows a lack of focus on research PM personal competence and development of its instructional blocks using Bloom's Taxonomy. Therefore, the study at hand aims to identify which personal competence elements as defined in the PMCD Framework are the undergraduate students less proficient in and how those competencies can be developed in the students.

## 3. Research Methodology

This study aims to answer the following questions: "In which area of personal competence do undergraduate-level project management students lack competence?" and "How can this project management competence be improved?"

This section aims to argue for and justify the methodological orientation and data analysis used in this research. This study falls in the positivist paradigm as it is empirical and reductionist, employs quantitative methods and findings are generalizable [47]. The methodology employed for this research constitutes a descriptive and (educational) action research approach with a quantitative mode of inquiry. The use of quantitative analysis is particularly appropriate as it helped reduce time and effort and generate a set of insights and propositions that addressed the critical elements involved in this research. As the notion of PM personal competence is relatively less practiced, this approach is being presented for graduate work readiness and its potential benefits to the graduates, as well as employers.

We followed a descriptive research method to better define the personal competencies held by the population. Frequency and percentage analysis techniques were applied to analyze the data collected from the students of different degree programs of an educational institute. These degree programs had Project Management as a core course in their curricula.

### 3.1. Data Collection and Analysis

To address the research objective, a survey instrument is used with cluster sampling to gain insight from the participants. The sample consists of undergraduate students enrolled in a mid-size university in Islamabad, Pakistan. Data collection for this study was done in a four-step process. It includes a survey based on PMCD performance criteria for Personal Competence, post-survey follow-up sessions to assess the responses to the first survey, and a second survey to further limit the problematic areas to one single area. The fourth step of data collection involving a focus group with Subject Matter Experts (SMEs) was done to check the validity of the content of the instructional block according to the findings of the previous steps.

3.1.1. First Step: Personal Competence Assessment (Survey 1)

In the first step of this research, the performance criteria mentioned for the third dimension of personal competence from the PMCD Framework by PMI are used as the basis for survey items. Probability Sampling, more specifically Cluster Sampling, is used to

select the sample. Each higher education institute is a cluster that is a mini-representation of our population as a whole. The reason behind the use of a survey was the large sample size of 149 students. The research population for this study included undergraduate PM students from Technical and Business education programs. For the sample to resemble the entire population as closely as possible, an entire batch of undergraduate students registered in Project Management courses from Electrical Engineering, Computer Sciences, and Business Administration programs is selected, i.e., all elements of an entire cluster are selected. The sample has no constraint on the gender of the students. Similarly, this research does not consider classifying responses according to the disciplines. The ages of the students ranged between 19 and 25 years. Survey responses were registered on a 5-point Likert scale. To analyze the collected data, Frequency Analysis was done using Microsoft Excel.

The first survey instrument was an elaborate questionnaire with a separate section for each of the six Units of the Personal Competence dimension of the PMCD Framework. The survey had one item against each performance criterion mentioned in the PMCD Framework and asked about the respondents' current level of 'proficiency' in each of the performance criteria for this dimension corresponding to multiple units of competence. This was done using the Project Manager Competency Summary Scorecard for developing competence as a template to find out the areas of significant gaps [48]. These units are:

1. "Achievement and Action" (30 items),
2. "Helping and Human Service" (13 items),
3. "Impact and Influence" (15 items),
4. "Managerial" (20 items),
5. "Cognitive" (7 items), and
6. "Personal Effectiveness" (14 items).

All six Units of Competence are further split into Competency Clusters, which then consist of several Elements comprising the Performance Criteria. Competency Clusters, corresponding Elements, and their respective Performance Criteria have been used as defined in the PMCD Framework.

To interpret and analyze the data, frequencies, percentages, and means/averages of responses were computed. Numeric values were assigned to the Likert scale, ranging from Never (1) to Always (5) so that lower computed averages in a performance criterion indicate that more students lack in that area. The lowest average shows that the "Performance Criteria" research population lacks the most.

### 3.1.2. Second Step: Post-Survey Follow-Up

Analysis of the results of the survey from the first step revealed very unexpected outcomes. This necessitated conducting the second step of data collection that comprised of three unique focus groups for the follow-up to discuss the reasons behind their responses. Three separate follow-up sessions were conducted with the same respondents to cater to their large number. Each session had about 50 respondents as participants.

Students were asked to express and discuss their views about personal competence elements.

The first couple of statements with the lowest average were selected to be discussed in the session. Respondents were asked about their grasp of the concepts that were being talked about in the statement and the reason for their counterintuitive responses. Their responses in focus groups led to the third step of conducting another survey about personal competencies.

### 3.1.3. Third Step: Personal Competence (Survey 2)

After the post-survey follow-ups, the third step of data collection began. Another survey was designed using both qualitative and quantitative judgment based on the results of the first survey and follow-ups. This second survey focused on the themes extracted from the top 7 problem areas as identified from the results of the first survey and the

follow-ups. Responses were registered in percentages. To analyze this survey, the means (averages) of the responses were calculated. A higher average percentage showed a greater lack of competence in that area.

A general idea of the main areas of Personal Competence that respondents lacked was conceived after the analysis of Survey 1. However, the serious lack of grasp on specific personal competence-related concepts seen in the follow-up sessions necessitated designing another survey. The second survey was designed by targeting broader Personal Competence areas with generic but focused statements to improve participants' understanding of competence areas. This was also done to further narrow down the identified areas of competence to one area where competence lacked the most.

From the top 7 items of Survey 1, eight themes were identified to develop a second, short, and focused survey. The same respondents were asked again about how much they thought the lack of these abilities affected their PM skills. Respondents were asked to specify their response in a percentage against every survey statement, where a higher percentage showed that respondents lack in that area more. The responses were tabulated, and the mean scores of the responses were computed against each problem area identified.

Once the key area in which the research population lacks competence was identified, an instructional block was tailored to meet their specific needs using Bloom's Taxonomy of Educational Objectives. Simmons et al. [49] report the benefits of inculcating professional competencies in undergraduate students to make them industry-ready. The content of that instructional block was used in the fourth step to be given to the SMEs for further vetting.

### 3.1.4. Fourth Step: Training Design, Evaluation, and Feedback

Subsequently, in the fourth step, training designed to develop the identified competence was thoroughly vetted by a focus group of five SMEs with vast experience, ranging from 20 to 35 years, in both academia and industry. After scrutiny, SMEs provided their feedback against the content design of the training.

### Training Design

Using Revised BT, LOs were constructed on the first three cognitive levels of the BT for the concepts extracted after the second survey.

Based on the constructed LOs, contents on the concepts to be covered and activities for effective instruction were selected in the context of the performance criteria of Personal Competence. Instructional blocks and the time allocated for each were decided with the consensus of SMEs. All subject experts agreed that the area identified needed much to be worked upon. A few weeks were given to focus group participants to scrutinize the training design, including outline, LOs, and the selected contents.

The experts were given a brief of complete research up until that point covering the introduction and purpose of the research. Details about the administration and results of Survey 1, its post-survey follow-up sessions, and Survey 2 were also shared with the experts.

### Training Evaluation and Feedbacks

To see if the trainees learn anything from the training, several methods can be used for evaluation, including Skill Assessment and Self-Report Measures. The self-report measure shows trainees' self-perception of their learning. To employ the self-report measure, a pre-training and post-training evaluation are designed that can be conducted to see how much improvement trainees believe to have in their competence. The evaluation form designed for this training contains statements related to the LOs with a 5-point Likert scale. To overcome the biases associated with self-report measures, another method of Skill Assessment can be used to test the participants about what they were taught. This test, designed using BT, contains questions from the activities that show if a concept has been grasped at the right level of BT. All questions are mapped against the LOs of the training.

Where evaluating the participants is necessary to see if they have learned anything, it is also important to appraise the trainer to see if they delivered the training effectively and assess if the concepts have indeed been covered in the training that it set out to cover. For this, a feedback form has been designed to measure to what degree the training contents covered the LOs and how effectively the trainer delivered the training. A 5-point Likert scale was used to record the responses on the feedback forms developed for both, training, and trainer evaluation.

## 4. Findings and Discussion

### 4.1. First Step: Personal Competence Assessment (Survey 1)

Table 1 shows the top 7 items of the first survey that had one item against each performance criterion of the Personal Competence dimension mentioned in the PMCD Framework. The statements are arranged in ascending order, showing the lowest scores for the items students are least proficient in:

**Table 1.** Survey 1: Personal Competence Assessment.

| Sr. No. | Performance Criteria Items | Averages |
|---|---|---|
| 1 | You: ["Disclose any possible conflict of interest."] | 3.34 |
| 2 | You: ["Act quickly and decisively in a crisis where the norm is to wait, and hope problem will resolve itself."] | 3.38 |
| 3 | You: ["Work independently and complete tasks without supervision."] | 3.38 |
| 4 | You: ["Use influential third parties to persuade others to support your actions, or to have a specific impact on the actions of others involved in the situation."] | 3.39 |
| 5 | In a professional setting, you can: ["Extend some contacts to informal or casual relationships—chats about family, interests, sports, news, and so on."] | 3.39 |
| 6 | You: ["Use stress-management techniques to control response, prevent burnout, and deal with ongoing stress, thus managing stress effectively."] | 3.46 |
| 7 | You: ["Drive increased effectiveness of the people."] | 3.50 |

The first statement of Table 1, "You disclose any possible conflict of interest." scored lowest with an average of 3.34. Such a low average with a staggering percentage of 21% of students responding that they 'seldom' or 'never' disclose a conflict of interest was alarming. Similarly, low scores for the subsequent statements showed that a further follow-up is required, as detailed in the next section.

### 4.2. Second Step: Post-Survey Follow-Up

To start, the first statement with the lowest average "you disclose any possible conflicts of interest" was selected to be discussed in the session. 21% of the total responses against this item about Conflict of Interest were 'never' or 'seldom'. Participants were inquired about their understanding of this concept and the reason why they did not disclose conflicts of interest.

The grasp of the concept by participants is shown in percentages in Table 2.

As tabulated in Table 2, in the first session, none of the respondents had the correct understanding of the concept as they mistook "Conflict of Interest" for general conflict management. Some answered that they would rather not reveal their internal conflicts to any irrelevant party or an outsider. Some respondents preferred to avoid the issue altogether and not confront it at all.

In the second session, only 20% and in the third session, only 39% of the total respondents had the right understanding of the concept 'conflict of interest', while the remaining respondents mistook it for conflict management and not revealing confidential information to an outside party. The results showed that out of all the participants of the follow-up sessions, only 16.5% had the right understanding.

**Table 2.** Post-survey Follow-up Sessions.

| Session | Understanding of the Concept | Percentage of Participants | |
|---|---|---|---|
| 1 | Wrong concept: Conflict Management | 100 | |
| | Will not disclose to any irrelevant party/outsider | | 92 |
| | Will not discuss at all | | 08 |
| 2 | Right concept | 20 | |
| | Wrong concept: | 80 | |
| | Conflict Management | | 20 |
| | Not to disclose any confidential information | | 60 |
| 3 | Right concept | 39 | |
| | Wrong concept: | 61 | |
| | Conflict Management | | 32 |
| | Not to disclose any confidential information | | 29 |

After knowing that participants of the follow-up had little or no grasp of the concept, another statement with a low average was picked for discussion. The statement "you do not hide or attempt to avoid conflict, but rather resolve it by bringing conflict out into the open" was about Conflict Management. A similar pattern of responses was observed for this statement as well. Participants in the follow-up session answered that they would rather not discuss the conflict with anyone—privy to it or an outsider. Such specific statements proved to be above their level due to the language barrier, little exposure, and the lack of practical experience of undergraduate scholars. They preferred to avoid confrontation. These unexpected responses led to the conducting of another survey about personal competencies, details of which are discussed hereafter.

*4.3. Personal Competence (Survey 2)*

From the top 7 performance criteria items of Survey 1 with the lowest averages given in Table 1, themes were identified to develop another survey. These themes were 'Professional Ethics', 'Communication', 'Conflict Management', 'Patience in Critical Situation', 'Power of Persuasion', 'Casual Attitude/Lack of Professionalism', 'Stress Management' and 'Motivating Others'. The same sample completed this second survey, the responses received were tabulated, and the mean percentage scores of the responses were calculated for each identified theme/problem area. Survey statements with a higher percentage showed an increased lack of proficiency in that area. The results in Table 3 show that the majority of the students identified the 'Communication' area as the one they lack the most.

**Table 3.** Survey 2: Personal Competence Assessment.

| Sr. No. | Statement | Percentage |
|---|---|---|
| 1 | "Lack of Communication Abilities" | 47.83 |
| 2 | "Lack of Stress Management Abilities" | 45.25 |
| 3 | "Lack of Patience in Critical Situations" | 42.61 |
| 4 | "Lack of Power of Persuading Others" | 41.61 |
| 5 | "Lack of Professionalism" | 40.98 |
| 6 | "Lack of Ability to Motivate Others" | 40.13 |
| 7 | "Lack of Conflict Management Abilities" | 40.01 |
| 8 | "Lack of Professional Ethics" | 36.86 |

Table 3 shows that most undergraduate students believe they need communication and stress management skills more than the other elements of personal competence. From a self-efficacy perspective, after considering which Personal Competence students think they lack the most, the top most competence 'Communication Skills' was identified. This is also reflected in the literature on what makes a successful project manager [11,50,51]. Khan [52] also establishes communication skills as the most important construct for employability. Simmons et al. [53] also report communication among the highest-ranked competencies. Podgórska et al. [54] and Yussof et al. [44] stress the crucial need to develop communication skills for high-complexity projects. Language proficiency has been identified as one of the most critical employability skills [55]. The results from Survey 2 were used to extract

some concepts that were related to communication skills from the Performance Criteria of Personal Competence as mentioned in the PMCD Framework. Communication skills were the one area research subjects lacked competence in the most. To enhance HEI students' Personal Competence, institution-led initiatives that focus more on practice are required. To develop the skills in this area, training was tailored to meet their specific needs using Bloom's Taxonomy of Educational Objectives, as discussed in the next section.

### 4.4. Training Design Using Bloom's Taxonomy

The concepts extracted after Survey-2 included "tailoring messages according to the audience", "networking and relationships", "ethical communication", "conflicts, negotiation, influence, and persuasion", and other specific situations in communication like "saying 'no'", "admitting mistakes", "delivering bad news", etc. It is important to note that a significant number of these skills are directly related to the capacity of the project manager to impact those on their team [56]. Hanif et al. [26] write that a deficit in formal training leads to knowledge gaps that undermine effective decision-making. Using Revised BT, LOs were constructed for these specific concepts on the three cognitive levels of the BT, as shown in Table 4. These LOs were specific enough to know what achievement is expected in a particular concept and, at the same time, broad and generic enough to have room for changes to the course and training content.

**Table 4.** Mapping of LOs against BT Levels and Concepts.

| Bloom's Cognitive Level | Sr. No. | LO Description | Concepts |
|---|---|---|---|
| Remembering | 1 | "The attendant should be able to identify and define the concept of communication and the key components of the communication process". | C1. Communication |
| Understanding | 2 | "The attendant should be able to demonstrate knowledge of communication concept, its process, and its applications; and to relate it to different situations". | C1. Communication |
| | 3 | "The attendant should be able to understand emotional intelligence and non-verbal communication, and why it is important to personal and professional success". | C7. Emotional Intelligence and Non-verbal Communication |
| Applying | 4 | "The attendant should be able to apply communication concepts and theories for the production of messages and to tailor them according to the situation and audience". | C2. Tailoring Message |
| | 5 | "The attendant should be able to apply concepts of communication to build and maintain healthy and effective relationships in various settings and contexts". | C3. Networking and Relationships |
| | 6 | "The attendant should be able to demonstrate positive, appropriate and professional ethical behavior in communication exchanges". | C4. Ethical Communication |
| | 7 | "The attendant should be able to demonstrate competency in skills related to conflicts, negotiation, and persuasive discourse". | C5. Conflicts, Negotiation, Influence, and Persuasion |
| | 8 | "The attendant should be able to use strategies for managing specific contexts for communication (saying 'no,' admitting mistakes, delivering bad news, etc.)". | C6. Specific Contexts in Communication |

### 4.4.1. Training Content and Focus Group

Some suggested contents related to the communication skills required for competent PM professionals and other related activities for effective instruction were selected. Topics, contents to be covered, roleplay scenarios used, and the time allocated for each were

decided with the agreement of all SMEs. All SMEs in the focus group agreed that good communication is an essential skill needed in the industry. A few weeks after sharing the complete package of background research, purpose, and details about the administration and results of Survey 1, its post-survey follow-up sessions, and Survey 2 with SMEs, they were called in a focus group to thoroughly vet the instructional block including outline, LOs, and the selected contents. One of the SMEs suggested that 'Ethical Communication' may also be covered. Another suggested that contents be simplified for the research population as English is not their first language and plays a part in lack of competence in the area. One of the experts in BT suggested that concepts may be merged and Concepts and LOs must be kept one-on-one so that they are easily measurable. It was also suggested that content may be localized so that it is easy to relate to.

Table 4 shows the mapping of Revised BT levels and the concepts of communication skills against the LOs after incorporating the suggestions of the SMEs. Nearly all topics covered had an easy-to-follow step-by-step approach.

The instructional block included some role-play scenarios and other engaging activities to keep the learning exercises interactive. This approximately six-hour-long training instructional block covered the topics in accord with the LOs and concepts behind them using BT, as shown in Table 5.

**Table 5.** Topics Covered in the Training Instructional Block.

| Sr. No. | Topics |
| --- | --- |
| 1 | Introduction to the training |
| 2 | Basics of Communication |
| 3 | Tailoring Message |
| |     Crafting the Written Message |
| |     Acquiring Great Presentation Skills |
| 4 | Networking and Relationships |
| |     Good Relationships |
| |     Networking Objectives and Network Map |
| |     Elevator Pitch |
| 5 | Ethical Communication |
| |     Principals of Ethical Communication |
| |     Grapevine Communication |
| |     Ethical Dilemmas |
| 6 | Conflicts, Negotiation, Influence, and Persuasion |
| |     Conflicts |
| |     The Conflict Layer Model |
| |     The Influence Model |
| |     The Persuasion Tools Model |
| 7 | Specific Contexts in Communication |
| |     Delivering the Bad News |
| |     Apology |
| |     Communicating Bad News to Customers |
| | Saying No |
| 8 | Emotional Intelligence and Non-verbal Communication |
| |     Emotional Intelligence |
| |     Non-verbal Communication |

The selection of the content selected in Table 5 was done keeping in mind the context of the PM Personal Competence dimension and its performance criteria. SMEs concurred that these contents best covered the communication skills required in a professional setting.

4.4.2. Training Evaluation and Feedbacks

To measure the effectiveness of the instructional block, Skill Assessment and Self-Report Measures can be employed for a pre-training and a post-training evaluation. The evaluation form designed for this training contains statements related to the LOs measured

on the Likert scale. Trainees can also be tested about what they were taught to overcome the bias of self-report measures. For a more thorough assessment, it is equally important to evaluate the trainer for effective delivery of the contents using a feedback form measured on a Likert scale.

In the PM curriculum, Knowledge and Performance Competencies are focused almost exclusively. Personal Competence is often not made a part of the curriculum. Undergraduate PM scholars show the most shortcomings in the Performance Criteria of Personal Competence that are related to Communication Skills, as mentioned in the PMCD Framework. By employing Bloom's Taxonomy, training can be planned and developed that is specifically tailor-made to cater to the immediate needs of the scholars. Training designed for Communication Skills should have a considerable focus on skill development instead of completely imparting knowledge. Proper mapping of concepts being covered to the LOs, as well as the levels of BT, should be ensured. An all-inclusive Training Design should have Training Assessments after, as well as before, the training, an exam for Skill Assessment, and Feedbacks of Training and Trainer accompanied by their analysis methods.

## 5. Conclusions

After reviewing the literature on PM competencies and related areas, we concluded that most of the research on PM competencies fails to address the development of key personal competencies required to be a competent project manager. The lack of a few crucial personal skills and competencies has become the reason for increased graduate unemployment. The work environment is continuously changing and getting increasingly competitive by the day, and the global job market is in an ever-bigger demand for skilled labor. The industry is continuously looking for graduates who are not just skilled in PM knowledge and performance aspects but in sociotechnical aspects as well. However, it is evident from the literature, the study of PM standards and baselines published by the professional bodies of Project Management that personal competencies have been neglected and not paid as much importance to as the technical aspects. Similarly, studies related to PM education also show that PM curricula largely focus on knowledge and performance competence. This leads to the need to bridge the gap between the requirements of the industry and the perceived requirements of academia. To bridge this gap, one must first find out which area of personal competence undergraduate scholars lack competence in and then how that competence be developed. While the findings of this study may not be generalizable to all HEIs and all undergraduate programs of different disciplines, they can serve as a step in the direction of understanding graduate employability and show how HEIs can cater to the work readiness of students for industry.

The findings of this study show that the majority of respondents lack understanding due to language barriers and have little exposure to the concepts. Students lack the essential PM personal skills and the understanding of the performance criteria. Follow-up focus group sessions and the second survey identify communication skills as the key problem area. Past studies with employers and the subject-matter experts of the current study who are from the industry concur with this finding. Instructional blocks designed and developed using Revised BT based on these findings, carefully scrutinized by the subject-matter experts, are all-inclusive and address the immediate needs of the students. The instructional block can be evaluated and assessed through the training and trainer feedback forms to supplement and prove its effectiveness. This research concludes that students were found to lack the essential Communication Skills required to be competent project managers. As is the case with the technical skills of the PM, Bloom's Taxonomy and Outcome-based Learning method must also be used to develop the instructional block for soft skills.

This research not only contributes a consolidated literature review on PM competence and graduate employability but also provides insight into the role of higher education institutes in underemployment. These findings have implications for HEIs, showing that it

is important to embed personal competence in the regular Project Management curriculum to improve the industry readiness of graduates.

## 6. Recommendations

Some of the core recommendations of this research are as follows:

1.  The significance of Personal Competencies must be recognized and focused on in PM courses in the future.
2.  The curriculum for this course must include training on much-needed soft skills, more specifically communication skills.
3.  More training on other Personal Competencies must also be designed and made a part of the PM curriculum.
4.  BT must be employed for any future training development to better the outcomes.
5.  The skill requirements of the students must be kept in view while creating the LOs and selecting the contents and activities using Bloom's Taxonomy Wheel.

## 7. Limitations and Future Work

The limitations of the study include data collection using a single source of questionnaires [57], use of self-report measures [58], participants from a single university [59], and a self-selection bias on the part of the selected sample and the researcher.

For future work to further the research in this area:

1.  Training based on the identified competence can be designed and conducted to improve the outcomes.
2.  The training developed in this research can be conducted to answer the following questions:

    *"How does competence-building training impact on research subject's project management abilities?"*

    Pre- and post-training assessments will be conducted to see the effect of training on the research subjects' abilities.

    relational hypothesis, *"Project manager personal competence development complements project management abilities of the undergraduate students,"* can be tested based on this.
3.  Best practices of other countries can be explored, and comparisons can be made to back the proposed competency-building technique.
4.  Independent analysis can be done for the different disciplines of the population under study, i.e., management (Business Administration) and technical (Electrical Engineering, Computer Sciences) disciplines. It can be assessed if the training content designed is equally applicable to different disciplines or if changes need to be made.
5.  It is important to note that there are other stakeholders in graduate employability and work readiness, such as employers, academicians, and graduates. Different stakeholders may hold different perspectives.

**Author Contributions:** Conceptualization, S.S. and A.A.; methodology, S.S. and A.A.; software, S.S.; validation, S.S. and A.A.; formal analysis, S.S. and A.A.; investigation, S.S.; resources, S.S., A.A., N.A. and O.H.; data curation, S.S. and A.A.; writing—original draft preparation, S.S., A.A. and N.A.; writing—review and editing, S.S., A.A., N.A. and O.H.; visualization, S.S.; supervision, S.S. and A.A.; project administration, S.S., A.A., N.A. and O.H. All authors have read and agreed to the published version of the manuscript.

**Funding:** This research received no external funding.

**Institutional Review Board Statement:** The study was approved by the Ethics Committee of Sir Syed-CASE-Institute of Technology (25 August 2019).

**Informed Consent Statement:** Informed consent was obtained from all subjects involved in the study.

**Data Availability Statement:** The data presented in this study are available on request from the corresponding author.

**Conflicts of Interest:** The authors declare no conflict of interest.

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
