# Peer review of "Students’ Workplace Readiness: Assessment and Skill-Building for Graduate Employability"

_sustainability, doi:10.3390/su14031749_

Round 1
Reviewer 1 Report
The paper deals with the development of socio-technical skills in the project management course taught in the undergraduate programs. The studies are oriented towards the identification of personal competencies, but also the elaboration of actions and educational material to form these competencies. Surveys, discussions with research subjects are used, and a skills training plan is indicated. The results of the study can be a support for increasing the employability of students.
The objectives of the study and the research methodology are clearly described.
I find some typos in identifying the authors – it is missing poz 3 affiliation.
The editing of the institutional members is deficient, the faculty / department are missing, the use of abbreviations is not recommended, and the characters must be uniform.
In lines 9 and 16 unbold “This”.
Other errors occur several times when inserting bibliographic references in the text: lines 356, 361, 374,375, 377, 398,405, 408, 430, 450, 456, 458, in which the expression “Delete Error should be replaced! Reference source not found.” should be replaced.
Also in lines 522-523 - delete one row.
Replace Ariticle with Article in line 1.
Overall, the paper has a structure suitable for research.
Author Response
|
Reviewer 1 |
||
|
Comments |
Responses |
Page Nos |
|
For empirical research, are the results clearly presented? |
Findings are more clearly presented. |
9-14 |
|
Comments and Suggestions for Authors |
|
|
|
I find some typos in identifying the authors – it is missing poz 3 affiliation. |
Poz 3 affiliation?
Missing department information added. Abbreviation changed to full form. |
|
|
The editing of the institutional members is deficient, the faculty / department are missing, the use of abbreviations is not recommended, and the characters must be uniform. |
1 |
|
|
In lines 9 and 16 unbold “This”. |
Couldn’t find these anywhere in the submitted manuscript copy (PDF) downloaded from the MDPI submission portal. No indicated formatting issues are found. All references are being shown. No indicated misspelling is found. |
|
|
Other errors occur several times when inserting bibliographic references in the text: lines 356, 361, 374,375, 377, 398,405, 408, 430, 450, 456, 458, in which the expression “Delete Error should be replaced! Reference source not found.” should be replaced. |
||
|
Also in lines 522-523 - delete one row. |
||
|
Replace Ariticle with Article in line 1. |
||
|
|
||
Reviewer 2 Report
- A more clear steps methodology and better justification of methods is needed. The method employed is not the most suitable
-A part of literature is dated and fails to engage with recent bibliography
- The Sample is scarce, is not very representative
- Findings need to be clear and make a clear contribution to theory, and fails in deep analysis of results presented.
- There is little interpretation as well as policy and managerial implications
- Cross check all references within text with your reference list. You may like to add more recent and relevant references published in recent months/years
Paper really needs a several reformulation and improvement in literature review, methodology and results.
Author Response
|
Reviewer 2 |
||
|
Comments |
Responses |
Page Nos |
|
Moderate English changes required |
Corrections are made. |
Throughout the manuscript. |
|
Is the content succinctly described and contextualized with respect to previous and present theoretical background and empirical research (if applicable) on the topic? |
Addressed. |
3-6 |
|
Are the research design, questions, hypotheses and methods clearly stated? |
More clearly stated. |
6-9 |
|
Are the arguments and discussion of findings coherent, balanced and compelling? |
Discussion of findings and results are more clearly presented. |
9-14 |
|
For empirical research, are the results clearly presented? |
||
|
Is the article adequately referenced? |
Improvements made. |
4-6 |
|
Comments and Suggestions for Authors |
|
|
|
A more clear steps methodology and better justification of methods is needed. The method employed is not the most suitable |
More clear and elaborate. |
6-9 |
|
A part of literature is dated and fails to engage with recent bibliography |
LR is updated. |
4-6, 12 |
|
The Sample is scarce, is not very representative |
Elaborated with justification. |
6 |
|
Findings need to be clear and make a clear contribution to theory, and fails in deep analysis of results presented. |
Findings are more clearly presented. |
9-14 |
|
There is little interpretation as well as policy and managerial implications |
Implications for HEIs added. |
15 |
|
Cross check all references within text with your reference list. You may like to add more recent and relevant references published in recent months/years |
Cross-checked all references. Added more recent references. |
4-6 |
|
Paper really needs a several reformulation and improvement in literature review, methodology and results. |
Improvements made. |
Various pages/ sections. |
Round 2
Reviewer 2 Report
the manuscript has been sufficiently improved to warrant publication in Sustainability